# Novel Polymorphisms in *RAPGEF6* Gene Associated with Egg-Laying Rate in Chinese Jing Hong Chicken using Genome-Wide SNP Scan

**DOI:** 10.3390/genes10050384

**Published:** 2019-05-20

**Authors:** Syed Ali Azmal, Ali Akbar Bhuiyan, Abdullah Ibne Omar, Shuai Ma, Chenghao Sun, Zhongdong Han, Meikuen Zhang, Shuhong Zhao, Shijun Li

**Affiliations:** 1Key Laboratory of Agricultural Animal Genetics, Breeding and Reproduction, Ministry of Education, College of Animal Science and Veterinary Medicine, Huazhong Agricultural University, Wuhan 430070, China; azmal@webmail.hzau.edu.cn (S.A.A.); aab@blri.gov.bd (A.A.B.); abdullah.ohi2014@webmail.hzau.edu.cn (A.I.O.); koma456@163.com (S.M.); shzhao@mail.hzau.edu.cn (S.Z.); 2Department of Livestock Services (DLS), Under the Ministry of Fisheries and Livestock (MOFL), Dhaka 1000, Bangladesh; 3Biotechnology Division, Bangladesh Livestock Research Institute, Under the Ministry of Fisheries and Livestock (MOFL), Dhaka 1000, Bangladesh; 4Huadu Yukou Poultry Industry Co. Ltd., Beijing 100000, China; chenghaosun567@gmail.com (C.S.); zhongdonghan82@gmail.com (Z.H.); 5DQY Ecological Co. Ltd., Beijing 100000, China; meikuenzhang@gmail.com

**Keywords:** Jing Hong chicken, egg-laying rate, GWAS, SNPs, *RAPGEF6*, association

## Abstract

The improvement of egg production is of vital importance in the chicken industry to maintain optimum output throughout the laying period. Because of the elongation of the egg-laying cycle, a drop in egg-laying rates in the late laying period has provoked great concern in the poultry industry. In this study, we calculated the egg-laying rate at weeks 61–69 (60 days) of Jing Hong chickens parent generation as the phenotype, and the genotype were detected by the chicken 600K Affymetrix Axiom High Density (HD) Single Nucleotide Polymorphisms (SNP)-array. The Genome-Wide Association Study (GWAS) result showed that the egg production trait is significantly associated with five SNPs (AX-75745366, AX-75745380, AX-75745340, AX-75745388, and AX-75745341), which are in the rap guanine nucleotide exchange factor 6 (*RAPGEF6*) gene on chicken chromosome 13. A total of 1676 Chinese commercial Jing Hong laying hens—including two populations, P1 population (858 hens) and P2 population (818 hens)—were genotyped using the Polymerase Chain Reaction-Restriction Fragments Length Polymorphisms (PCR-RFLP) method for the association analysis of egg-laying rates for the verification of the GWAS results. Genotypic and allelic frequencies of five SNPs were inconsistent with Hardy–Weinberg equilibrium, and the average population genetics parameters considering all the SNP values; i.e., gene homozygosity (*Ho*), gene heterozygosity (*He*), the effective number of alleles (*Ne*), and the polymorphism information content (*PIC*) were 0.75, 0.25, 1.40, and 0.20 in P1; 0.71, 0.29, 1.46, and 0.24 in P2; and 0.73, 0.27, 1.43, and 0.22 in P1 + P2 populations, respectively. The association analysis results revealed that out of the five polymorphisms, three of them (AX-75745366, AX-75745340, and AX-75745341; Patent applying No: 201810428916.5) had highly significant effects on egg-laying rates according to the GWAS results. Population-specific association analyses also showed similar significant association effects with this trait. Four haplotypes (AAGG, AAAG, AGGG, and AGAG) were inferred based on significant loci (AX-75745340 and AX-75745341) and also showed significant associations with the egg-laying rate, where haplotype AAGG had the highest egg-laying rate, with the exception of the egg-laying rate in P1 population, followed by other haplotypes. Furthermore, genotypes TT, AA, and GG showed the highest egg-laying rate compared to the corresponding genotypes at AX-75745366, AX-75745340, and AX-75745341 SNP loci in P1+P2, respectively. A similar result was found in the population-specific analysis except for the P1 population, in which TC genotype showed the highest egg-laying rate. No significant association was found in the egg-laying rate during the 60 days laying period for the SNPs (AX-75745380 and AX-75745388) in any group of population (*p* ≥ 0.05). Collectively, we report for the first time that 3 SNPs in the *RAPGEF6* gene were significantly associated with the egg-laying rate during the later stage of egg production, which could be used as the potential candidate molecular genetic markers that would be able to facilitate in the selection and improvement of egg production traits through chicken breeding.

## 1. Introduction

Chicken eggs are rich in protein, fatty acids, vitamins, and minerals, and are measured as an excellent source of animal protein. With the enhancement of laying hens’ production performance, the laying age of commercial laying hens has been prolonged from the first 72–80 weeks, and some breeding companies have even-continued the laying cycle to 100 weeks, recommending the breeding program “breeding for 500 eggs in 100 weeks” [1,2,3]. However, the rapid drop in egg-laying rates at the end of the laying cycle has severely hindered the attainment of this goal. Therefore, improving the egg-laying rate plays a vital role in realizing this program and extending the laying cycle. Improvement of egg production is a matter of concern in the poultry industry to enable egg producers to meet the enormous global demand. It is essential to maintain the optimum level of production throughout the laying period to make poultry farming profitable by minimizing the cost of production [4,5,6]. Published reports suggest that some birds in a layer flock should target the level of egg production at even the later stage of the rearing period; however, some laying hens have failed to achieve the expected production performance. Under such circumstances, the selection of layer birds showing a better egg production performance at a later stage may make layer farming significantly more profitable. During the late period of production, some laying chickens lay eggs at a productivity rate of 95% of maximum productivity, whereas some laying chicken lays eggs less at a productivity rate of than 80%, which why the selection of higher egg-producing hens by breeding can make more benefit. The egg-laying performance in a particular layer flock fluctuates over time and can be represented in terms of a “production curve”. The nature of the curvature is demarcated by subsequent stages, such as (a) sexual development (which results from the onset of production), followed by the highest production or (b) maximum egg production, followed by a slow decline in egg production or (c) continuation of egg production. Traditional breeding strategies designed to improve the efficiency of chicken egg production are based on a long term monitoring of egg numbers and laying rates; however, such techniques are always time-consuming [7]. During peak production, hen-day egg production in modern layers almost reaches its highest biological potential, attaining 96.98% of production (i.e., one egg per layer everyday); resulting in barely any dissimilarities in egg production performance among birds at this stage. Whatever fluctuates among layers is how extended they can continue a high rate of lay and at what rate production decreases after the peak [8,9,10]. Continuing a persistency of laying in the late period may have a positive impact on a flock’s financial performance. An inherently augmented persistency in egg production of commercial layer enhances its performance to keep flocks at a more extended period of production.

A good number of previous investigations have been identified several polymorphisms in different genes of laying chicken that are closely associated with egg production traits including egg number, laying rate, egg weight, and egg quality characteristics [5,6,9,10,11,12,13,14,15,16,17,18,19,20]. The identification of a quantitative trait locus (QTL) controlling chicken reproductive performances for solicitation in marker-assisted selection (MAS) has been progressing quickly [21,22]. Reproduction traits are controlled by QTL, and a genome-wide scan is a practical approach that can be used to gain an understanding of these complex traits. As a statistical tool, a Genome-Wide Association Study (GWAS) is one of the most effective methods for identifying essential Single Nucleotide Polymorphisms (SNPs) and functional genes that affect quantitative traits [23]. The technique is more efficient at identifying genetic characteristics for economic traits than the candidate gene approach. In the present study, a GWAS was performed on 120 Jing Hong chickens of parent generations for 60 days at 61–69 weeks of age to observe egg production performance as the phenotype. Further, the genotype was detected by the chicken 600K Affymetrix Axiom HD SNP-array to identify molecular markers and candidate genes associated with the egg-laying rate. The results found that five SNPs in chromosome 13 were significantly associated with the egg-laying rate. However, the GWAS usually focuses on common variants only, because genetic markers (SNPs) that show minor allele frequencies, lower than 0.05, are generally excluded from the analysis.

The rap guanine nucleotide exchange factor 6 (*RAPGEF6*) is a protein-coding gene located on chromosome 13 in chickens (Gene ID: 101751943). Gene Ontology (GO) annotations associated with this gene consists of a guanyl-nucleotide exchange factor movement and a GTP-dependent protein requisite. The nucleotide sequence (15794705-15904202) specifies that this gene consists of 33 exons that encoded 109,498 amino acids (Accession No. NC_006100.3). The *RAPGEF6* gene is preserved in transversely species comprising humans, chimpanzees, rhesus monkeys, dogs, cows, mice, rats, zebrafish, and *Caenorhabditis elegans*. Rap1, a member of the protein of the Ras-like superfamily of small GTPases, is involved in maintaining widespread cellular functions linked to cell adhesion, proliferation, differentiation, spreading, and endothelial junction control [24,25,26,27,28,29,30]. The accumulating results indicated that the *RAPGEF6* gene has been established to perform a fundamental role in spermatogenesis [31,32], schizophrenia [33], and neuro psychiatric disorders [34] in mice, indicating that *RAPGEF6* is essential for the reproductive development in mice. No polymorphisms in the *RAPGEF6* gene that may affect economic traits in chicken have been investigated. Therefore, this study was planned to examine SNPs in the *RAPGEF6* gene and their associations with the egg production trait to evaluate the effect of this gene on the egg-laying rate in chicken. This will provide further useful and detailed information that can be used to advance poultry breeding through the use of molecular marker-assisted selection programs.

## 2. Materials and Methods

### 2.1. Ethics Statement

This research was performed in strict accordance with the guidelines for experimental animals established by the Ministry of Science and Technology of China. All experimental procedures and research on animals were conducted in strict conformity with the recommendations in the Guide for the Care and Use of Laboratory Animals according to the regulations proclaimed by the Standing Committee of Hubei People’s Congress (No. 5) and approved by the Biological Studies Animal Care Committee of Hubei Province, China as well as the ethics committee of Huazhong Agricultural University, Wuhan, China (Permission number: 4200896859).

### 2.2. Experimental Birds

A total of 1676 Chinese Jing Hong laying chicken from two populations were used for genotyping and marker-trait association analysis under this study. The first population (P1), comprising 858 healthy Chinese Jing Hong laying chickens (61–69 weeks old), were obtained from the poultry farm Jingzhou Yukou Poultry Industry Co. Ltd., Jingzhou-434020, Hubei, China. The other population (P2), comprising a total of 818 Chinese Jing Hong laying chickens (also 61–69 weeks old), were reared at Huadu Yukou Poultry Industry Co. Ltd., Beijing, China. These birds were raised individually in stair-step cages under a consistent environment with separated feed trays and egg collecting trays. The birds were fed by limited nutrition according to the percentage of their weights. The birds were fed a commercial corn-soybean-based iso-caloric and iso-nitrogenous diet, containing 2850 Kcal ME/kg dry matter (DM) and 16.83% crude protein (CP). The required feed was supplied twice a day, at 7:00 a.m. and at 4:00 p.m., according to the Chinese Jing Hong laying chicken rearing manual. Safe drinking water was also made available at all times by a nipple drinker. All birds were kept in identical light/dark cycles that exposed them to 16 h of light per day.

### 2.3. Data Collection and Measured Traits

The number of eggs produced from the individual experimental bird was recorded daily for 60 days duration (weeks 61–69). The egg-laying rate was calculated from data concerning the number of eggs produced by each hen using the following equation:
$$\mathrm{Egg}-\mathrm{Laying}\;\mathrm{rate}\;\left(\mathrm{LR}\right)\%=\;\frac{\mathrm{Total}\;\mathrm{number}\;\mathrm{of}\;\mathrm{eggs}\;\mathrm{produced}}{\mathrm{Days}\;\mathrm{of}\;\mathrm{production}}\;\times \;100$$

To construct the evolutionary relationships of the *RAPGEF6* gene of chicken with other species, we also downloaded nucleotide sequences of different species from the National Centre Biotechnology Information (NCBI) (Rockville Pike Bethesda, MD, USA) and kept chicken as a reference. All collected sequences and their corresponding nucleotide sequences were aligned, and a phylogenetic tree was constructed using Molecular Evolutionary Genetics Analysis (MEGA) version 6 software with the maximum likelihood method [35]. The analysis results of the chickens’ *RAPGEF6* gene were compared with 16 species and the genetic similarity results were recorded.

### 2.4. Blood Sample Collection and Genomic DNA Extraction

Blood samples were collected from the wing vein in a tube containing ethylene diamine tetra acetic acid (EDTA), as an anticoagulant. All samples were collected in an ice box and subsequently preserved at −20 °C until further use. The genomic DNA was extracted according to the phenol-chloroform method as described by Sambrook and Russel, 2006 [36], with some minor modifications. The concentration and quality of the extracted DNA were quantified using the ND-2000 spectrophotometer (NanoDrop Co., Ltd., Thermo Fisher Scientific, Madison, WI, USA) and agarose gel electrophoreses, respectively. Only genomic DNA preparations within the ratios of 1.6–1.8 (A260/A280) and equilibrated to 50 ng/μL were used for the genotyping.

### 2.5. Identification of SNPs Associated with Egg Production Traits Using GWAS

Jing Hong parent hens showing extreme performance differences were used to detect SNP markers associated with egg-laying rates using the GWAS. A high-density SNP array was employed here to identify associated variants underlying egg production traits using the chicken 600K Affymetrix Axiom HD SNP-array (Aviagen Ltd., Midlothian, UK) to investigate whether the effects of these QTLs were associated with egg production traits [37]. The GWAS was performed using a total of 120 Jing Hong parent chickens from a total of 858 hens of the first population. The chickens of the first sixty and of the last sixty egg production traits were used for the GWAS. The first sixty and the last sixty were considered as control chickens and case group chickens, respectively. Two statistical models-a fixed-effect linear regression model and a mixed-effect linear model—were used to estimate the association effects of SNPs on each of the phenotypes. The significant and suggestive thresholds were set at (*p* = 2.09 × 10^−7^) and (*p* = 4.18 × 10^−6^), respectively. The two statistical models concurrently identified five genome-wide significant SNPs (*p* < 0.05) on egg production traits in these chickens. The global view of *p*-values for all SNP markers was visualized using a Manhattan and Quantile-Quantile (Q-Q) plot that was constructed for each trait using the “CM plot” package in the R software (version 3.5) [38]. A GWAS analysis was performed using the compressed mixed linear model [39] carried out using the TASSEL software package (version 5.0) (Ithaca, NY, USA) [40]. The genome-wide significance *p* value threshold was determined by the “LD-adjusted” Bonferroni method [41]. Those five SNPs are all located in the *RAPGEF6* gene.

### 2.6. Genotyping by PCR-RFLP and Reconstruction of Haplotypes

All experimental birds (P1 and P2) were then genotyped considering the identified SNP’s location following the PCR-RFLP technique. A total of five primers were designed for the genotyping of 5 tag SNPs in the *RAPGEF6* gene. Tabular description of five identified Database SNPs in the *RAPGEF6* gene is shown in Appendix A. The *RAPGEF6* gene (Gene Bank Accession No. NC_006100.3, Region: (15794705-15904202) reference sequence was used to design primers to amplify DNA fragments that contain those SNPs with the Oligo6 software. Primer synthesis was completed using Oligonucleotide synthesis technology of Sangon Biotech Co., Shanghai, China. A detailed description of primer sets and their corresponding products size with the location is that were used to genotype five tag SNPs is shown in Appendix A. The 10.0 µL reaction mixture volume included 1.3 µL of the DNA template, 0.15 µL of each primer (100 nmol μL^−1^), 10 × buffer 1.0 µL, dNTP 0.1 µL, rTaq 0.1 µL, and 7.2 µL sterile distilled water used for PCR. The amplification program consisted of an Eppendorf thermal cycler that was programmed for an initial incubation temperature at 95 °C for 5 min, followed by 35 cycles each of which included denaturing at 95 °C for 30 s, annealing at 50–55 °C (Appendix A) for 30 s, and an extension at 72 °C for 40 s. A final extension at 72 °C for 7 min and then at 25 °C for 30 s were carried out. After the reaction process was completed, the PCR products were separated on 1.5% (w/v) agarose gel electrophoresis and stained with ethidium bromide and visualized in a BIO-RAD Image Lab gel documentation system (BIO-RAD, Hercules, CA, USA). The amplified PCR products included were digested by 0.03 µL of 250 specific restriction enzymes (New England BioLabs Inc., Ipswich, MA, USA). The digestion mixture contained: 5 µL of PCR products, 1 µL of cut smart buffer, 4 µL of DNAse free ddH_2_O, and 3.0 U of each enzyme. After that, it was incubated overnight at the manufacturer’s instructed temperature for specific restriction enzyme’s requirements. The genotype patterns were visualized from the digested products that were separated on 5% (w/v) agarose gel electrophoresis and stained with ethidium bromide. The genotype pattern was recorded by the gel documentation system (BIO-RAD Image Lab) to check the genotype category of each chicken considering each SNP’s position. The ingredients of SNPs in the *RAPGEF6* gene PCR-RFLP optimized Restriction Enzyme (*RE*) digestion mixture is shown in Appendix A. The restriction enzymes used in the present study and their respective restriction sites are given in Appendix A. Haplotypes were reconstructed among three significant SNPs (AX-75745366, AX-75745340, and AX-75745341) and association analysis of haplotypes were also performed based on two significant SNPs (AX-75745340 and AX-75745341) according to the genotyping data obtained from all experimental population of P1 and P2 used in the present study applying the PHASE 2.0 program [42]. The minimum haplotype frequency was set at 2%.

### 2.7. Polymorphism Evaluation

The genotype and allele frequencies at each SNP site were calculated for all populations according to electrophoresis results of the genotype categorization. Genotypic frequencies of different PCR-RFLP patterns were estimated from the combination of different RFLP alleles generated based on the presence or absence of one or more restriction sites. Different genotypes were identified based on different patterns. Individual Jing Hong layer chickens were genotyped for the five SNPs at base positions of 15836649, 15843452, 15829057, 15845449, and 15829303 and termed SNPs; SNP-01 (T-15836649-C), SNP-02 (C-15843452-T), SNP-03 (A-15829057-G), SNP-04 (G-15845449-A), and SNP-05 (A-15829303-G). Genotype frequencies of the SNPs in the *RAPGEF6* gene were analyzed by Microsoft Excel (MS Excel). Gene frequencies were calculated from genotypic frequencies. The test for the Hardy-Weinberg–equilibrium at each SNP site was conducted separately for the P1, P2, and P1 + P2 population, using the Haplo view version 4.2 software (http://www.broad.mit.edu/mpg/haploview/) [43]. Genotypic and allelic frequencies at each SNP site were calculated, with each polymorphism evaluated for the Hardy–Weinberg equilibrium using a Pearson’s goodness-of-fit chi-square test (degree of freedom = 1). We performed linkage disequilibrium (LD) analysis in order to characterize five causal SNPs in a strong LD region where three significant SNPs were identified by the solid spine algorithm in Haploview version 4.2 (Cambridge, MA, USA) as being clustered [43]. Gene homozygosity (*Ho*), heterozygosity (*He*), the effective number of alleles (*Ne*), and the polymorphism information content *(PIC)* were statistically analyzed using the POPGENE v. 1.32 software [44].

### 2.8. Marker-Trait Association Analysis

Association analyses of the five SNPs genotypes or haplotypes with Phenotypic data of egg-laying rate were performed in the Chinese Jing Hong layer chicken population using the General Linear Model Procedures (GLM) of the SAS statistical 9.2 software package (SAS Institute, Inc., Cary, NC, USA) [45]. Data were processed in MS Excel, and a linear mixed effect model procedure analyzed the genetic effects.
Y_ijk_ = μ + L_i_ + G_j_ + F_k_ + e_ijk_
where Y_ijk_ is phenotypic value of the target trait (e.g., egg-laying rate), μ is population mean of egg- laying rate, L_i_ is fixed effect of the line, G_j_ is fixed effect of the SNP genotype or haplotype, F_k_ is random effect of the family, and e_ijk_ is the overall error term.

*Type III* sum of squares was used in each test. The threshold for significance was set at *p* < 0.05 and for high significance at *p* < 0.01. All values are presented as least square means with standard errors of the mean (LSM ± SEM). Means were compared for significant differences using Duncan’s Multiple Range Test [46].

## 3. Results

### 3.1. Identification of Most Significant SNPs in *RAPGEF6* Gene by GWAS 

The global view of *p* values for all SNP markers was visualized by a Manhattan plot, as shown in Figure 1. After performing a GWAS, there was no dramatic deviation between observed and expected −log10 (*p* value) in the Q-Q (Quantile-Quantile) plot (Figure 2), suggesting that there was little or no evidence of residual population structure effects in test statistic inflation. GWAS results of 120 laying birds show that five identified SNPs in chromosome 13 are associated with egg production traits, which are all located in gene *RAPGEF6*. This gene could serve as a new candidate gene for the egg-laying rate during the late period of laying, yet their roles need to be verified in further studies. The positions of the SNPs and information were obtained based on the ICGSC annotation of the *Gallus gallus* genome version 4.0 and the gene within 109,498 (15794705-15904202) base pairs flanking the associated SNPs that were chosen for analysis. A tabular description of five identified database SNPs in the *RAPGEF6* gene is shown in Appendix A. The selected significantly associated SNPs by GWAS were further genotyped for validation using an PCR-RFLP approach, and association studies were performed with egg-laying rate trait in two expanded populations.

### 3.2. Phylogenetic Analysis

The nucleotide sequences were analyzed for the phylogenetic tree using MEGA version 6 software. The nucleotide sequences of the *RAPGEP6* gene of chicken (*Gallus gallus*), t emperor penguin (*Aptenodytes forsteri*), Adelie penguins (*Pygoscelis adeliae*), Golden Eagle (*Aquila chrysaetos* canadensis), bald eagle (*Haliaeetus leucocephalus*), common pigeon (*Columba livia*), swan goose (*Anser cygnoides* domesticus), mallard duck (*Anas platyrhynchos*), society (or Bengalese) finches (*Lonchura striata* domestica), hooded crow (*Corvus cornix* cornix), american crow (*Corvus brachyrhynchos*), ground tit (*Pseudopodoces humilis*), eurasian blue tit (*Cyanistes caeruleus*), great tit (*Parus major*), Japanese quail (*Coturnix japonica*), melmeted guineafowl (*Numida meleagris*), and wild turkey (*Meleagris gallopavo*) were aligned, and a phylogenetic tree was constructed to know the evolutionary relationship with other species using MEGA version 6 software (Pennsylvania State University, PA, USA) with the Unweighted Pair Group Method with Arithmetic Mean (UPGMA) method. The analysis results of the chicken *RAPGEF6* gene were compared with 16 species, and the genetic similarity results were recorded. The UPGMA phylogenetic tree showed that the chicken *RAPGEP6* gene was closely related to the *RAPGEP6* gene of the guineafowl (*Numida meleagris*) than that of other studied species (Figure 3).

### 3.3. Genotyping by PCR-RFLP and Reconstruction of Haplotypes

The genotype patterns of the SNPs in the *RAPGEF6* gene were checked using the PCR-RFLP technique and found three genotype patterns in SNP AX-75745366 (TT, TC, and CC) and two genotype pattern in SNP AX-75745380 (CC and CT), AX-75745340 (AA and AG), AX-75745388 (GG and AA) and AX-75745341 (AG and GG) (Appendix A). The genotypic and allelic frequencies of each identified SNP in the *RAPGEF6* gene are presented in Table 1. Haplotypes were reconstructed based on three significant SNPs-SNP-01: AX-75745366 (T15836649C), SNP-03: AX-75745340 (A15829057G), and SNP-05: AX-75745341 (A15829303G)-and their frequencies among all studied individuals are shown in Table 2. A total of eight haplotypes were constructed, in which the most abundant three haplotypes were *H1* (TAG), *H2* (CAG), and *H3* (TAA), which accounted for 94.77, 86.06, and 90.55% of the genetic information in P1, P2, and P1 + P2 populations, respectively. Haplotype *H1* (TAG) was predominant in all groups of the population studied. Another haplotype reconstruction was performed based on two significant SNPs-SNP-03: AX-75745340 (A15829057G) and SNP-05: AX-75745341 (A15829303G)-according to these genotyping data, and four haplotypes (AAGG, AAAG, AGGG, and AGAG) were identified among the individuals. For the P1 population, the haplotype present at the highest frequency was the AAGG haplotype (0.84), with the AAAG haplotype is the next most frequent (0.07), followed by AGAG (0.06), and AGGG haplotype is the lowest frequent (0.03). For the P2 population, the haplotype present at the highest frequency was the AAGG haplotype (0.65), with the AGAG haplotype being the next most frequent (0.14), followed by AAAG (0.13), and finally the AGGG haplotype as the lowest frequent (0.08). For the P1 + P2 population, the haplotype present at the highest frequency was the AAGG haplotype (0.75), with the AAAG haplotype is the next most frequent (0.1), followed by AGAG (0.09), and AGGG haplotype is the lowest (0.06).

### 3.4. Frequencies of Genotypes and Alleles at the SNP Locus

The genotypic and allelic frequencies of each identified SNP in the *RAPGEF6* gene are presented in Table 1. For the SNP AX-75745366 the frequency of allele T was higher than allele C, with the frequency of genotype TC being higher than genotype TT and CC in all populations. For the SNP AX-75745380, the frequency of allele C was notably higher than allele T, with the frequency of genotype CC being predominantly higher than genotype CT in all population. For the SNP AX-75745340, the frequency of allele A was notably higher than allele G, with the frequency of genotype AA being predominantly higher than genotype AG in all populations. For the SNP AX-75745388, the frequency of allele G was higher than allele A, with the frequency of genotype GA being higher than genotype GG in all populations. For the SNP AX-75745341, the frequency of allele G was notably higher than allele A, with the frequency of genotype GG being predominantly higher than genotype AG in all populations. The whole population was found to exhibit significant genetic disequilibrium (*p* ≥ 0.05), with the exception of the P1 population in the SNP site AX-75745340 and AX-75745341 between concern alleles in the *RAPGEF6* gene. This shows a low genetic diversity in the population, which might result from the cause of selection. As shown in Table 3, the gene homozygosity (*Ho)* was higher than gene heterozygosity (*He)* for all of the locus as well as for all of the population, with the effective allele numbers (*Ne).* The value of polymorphism information content (*PIC)* was not higher for all of the locus as well as for all of the population. The Hardy–Weinberg equilibrium and the average population genetics parameters considering all the SNPs (i.e., *Ho*, *He, Ne* and *PIC* values) were 0.75, 0.25, 1.4, and 0.2 in P1; 0.71, 0.29, 1.46, and 0.24 in P2; and 0.73, 0.27, 1.43, and 0.22 in P1 + P2 populations, respectively. The *PIC* was not higher and the value of *PIC* for *He* in all groups of the population were 0.1074, 0.2077, and 0.1606, respectively. 

### 3.5. Association Analysis between the SNP Genotypes in *RAPGEF6* Gene with Egg-Laying Performance in Jing Hong Hens Breed

Statistical analyses were performed to test the significance of the difference of genotype effect on egg production performance among the five SNPs of *RAPGEF6* gene in Chinese local Jing Hong layer chickens. The association analysis results were shown between the SNP genotypes in the *RAPGEF6* gene and egg-laying rate (LR) by 60-day laying period in the Chinese Jing Hong layer chickens, and the least square means and standard error of means (LSM ± SEM) of different genotypes for each SNPs are listed in Table 4. The results showed that the significant (*p* < 0.0001) association with 60 days egg-laying rate were found at the loci AX-75745366 (T15836649C), AX-75745340 (A15829057G), and AX-75745341 (A15829303G) in all populations of Chinese local Jing Hong layer chicken, which are treated as SNP-01, SNP-03, and SNP-05 respectively. The SNP-01, SNP-03, and SNP-05 was significantly associated with the 60 days of egg production. In the SNP AX-75745366 (T15836649C), the significantly highest (*p* < 0.0001) egg-laying rate was shown in chickens with genotypes TC (80.50 ± 0.86), followed by the genotypes TT (78.63 ± 1.08), and were the lowest with genotypes CC (72.61 ± 1.62) in P1 population. The genotypes TT (85.30 ± 1.02 and 81.86 ± 0.75) showed the highest egg-laying rate in P2 and P1 + P2 population, followed by the genotype TC (80.49 ± 0.79 and 80.49 ± 0.59), and was the lowest with genotypes CC (76.40 ± 1.54 and 74.42 ± 1.13), respectively. In the SNP AX-75745340 (A15829057G), the significantly higher (*p* < 0.0001) egg-laying rate was found in chickens with the genotypes AA (79.99 ± 0.64, 83.28 ± 0.65, and 81.47 ± 0.46) in P1, P2, and P1 + P2 population compared with the genotype AG (65.69 ± 2.06, 74.87 ± 1.24, and 72.10 ± 1.09), respectively. In the SNP AX-75745341 (A15829303G), the significantly highest (*p* < 0.0001) numbers of egg production were seen in chickens with the genotypes GG (79.59 ± 0.66, 84.09 ± 0.67, and 81.57 ± 0.47) in P1, P2, and P1+P2 population compared with genotype AG (72.43 ± 1.79, 74.50 ± 1.09, and 73.84 ± 0.96), respectively. There was no significant association for the SNP AX-75745380 (C15843452T) and AX-75745388 (G15845449A) for the egg-laying rate for the studied 60-day laying period (*p* ≥ 0.05).

An association analysis between the haplotypes (AAGG, AAAG, AGGG, and AGAG) inferred based on two significant loci—AX-75745340 (A15829057G) and AX-75745341 (A15829303G)—and the egg-laying rate by day 60 in P1, P2, and P1 + P2 population are shown in Table 5. In the case of P1, among the haplotypes, haplotype AAAG was found to be correlated with the significantly highest (*p* < 0.0001) egg-laying rate (83.03 ± 2.33) for the 60-day laying period, followed by haplotype AAGG (79.79 ± 0.40) and AGGG (75.85 ± 3.24); the lowest levels were associated with haplotype AGAG (59.07 ± 2.62). However, in case of P2, among the four haplotypes, haplotype AAGG was shown to be markedly associated with the significantly highest (*p* < 0.0001) egg-laying rate (84.50 ± 0.70) for the 60-day laying period, followed by haplotype AGGG (80.46 ± 2.07) and AAAG (77.27 ± 1.56); the lowest levels were associated with haplotype AGAG (71.90 ± 1.51). Finally, in case of P1+P2 population, among the four haplotypes, haplotype AAGG was found to be correlated with the significantly highest (*p* < 0.0001) egg-laying rate (81.76 ± 0.49) for the 60-day laying period, followed by haplotype AAAG (79.28 ± 1.34) and AGGG (78.94 ± 1.81); the lowest levels were associated with haplotype AGAG (68.23 ± 1.36).

### 3.6. Linkage Disequilibrium (LD) Analysis of SNPs in *RAPGEF6* Gene in Chinese Jing Hong Chicken Population

Haplotype block and LD structures were generated from the five SNPs genotyped in the *RAPGEF6* gene from chicken populations (Figure 4a–c). Pairwise coefficients of linkage disequilibrium (D’) values are shown between polymorphisms, which were calculated from the genotypic data of P1 = 858, P2 = 818, and P1 + P2 = 1676 chickens. The haplotypes block was defined by using the default setting of the Haploview software (Broad Institute, Cambridge, MA, USA). In the P1 population, there were two variants: AX-75745341A/G and AX-75745341A/G showed non-significant LD with each other with low D’ (D’ = 13) and spanning 0 kb in only one block 1. In the case of P2 population, there were three variants: AX-75745340A/G, AX-75745341A/G, and AX-75745366T/C showed significant LD with each other with highly strong D’ (D’ = 93 and 83) and spanning 7 kb in block 1 only. Finally, in the case of the P1 + P2 population, two variants were found: AX-75745340A/G and AX-75735341/G showed significant LD with each other with strong D’ (D’ = 69) and spanning 0 kb in block 1 only.

## 4. Discussion

Egg-laying rates are an important economic trait. With the extension of the laying cycle, the drop in egg-laying rates during the late laying period, has provoked great concern [2]. The present research was designed to uncover the critical SNPs in the *RAPGEF6* gene that affect the egg-laying rate using egg production data from Jing Hong chickens with laying ages of 61 to 69 weeks. This study is the first to conduct a GWAS of egg-laying rates at the late laying period with the chicken 600 K high-density SNP array. Two populations of Jing Hong layer chickens were employed in this study to verify the GWAS results. Egg-laying performance is one of the most important economic traits in the poultry industry which is a crucial goal of breeding programs and has been attracting increasing interest. However, most of the egg production traits in chickens are inherited polygenically with low to moderate heritability [47,48,49]. Moreover, they are obtained only from sexually mature females, which makes genetic improvements in males more difficult to estimate using traditional methods depending on breeding value assessment. Innovatively improving and utilizing the egg production trait is becoming one of the essential tasks in the chicken industry. We tried to elucidate the relationships between polymorphisms of the *RAPGEF6* gene and egg production traits to identify potential candidate molecular genetic markers as an aid in the improvement of egg production through marker-assisted selection and breeding in chicken. Nevertheless, since egg production traits are polygenically inherited, more associated target genes and favorable alleles are required for the improvement of egg production performance. Therefore, the polymorphisms in the *RAPGEF6* gene were detected by the GWAS, screened using the PCR-RFLP method, and used to study genetic associations with egg production traits.

The genetic factors are destined to play a pivotal role in promoting egg production traits to further improve this economically important resource. In addition to a large number of the members of the hypothalamic–pituitary–gonadal hormone (HPG) axis, a wide variety of local intra-ovarian factors have been shown to play a critical role in normal follicular development and oocyte maturation, including the *RAPGEF6* gene [31,50,51,52,53,54]. Involved in the process of egg production are not only members of the glycoprotein hormone family of gonadotropins, such as follicle-stimulating hormone (FSH) and luteinizing hormone (LH), but also a wide variety of local intra-ovarian cellular and tissue-level signal transductions that play a critical role in regulating normal follicular development and oocyte maturation [11,12,21]. These processes are also controlled by many local intra-ovarian factors in an autocrine or/and paracrine manner, such as the α1B ADR and PGC-1β proteins [51,55,56]. Polymorphisms associated with egg production related hormones, growth factors, and sex hormones such as follicle-stimulating hormone beta subunit (FSHb), LH, prolactin (PRL), growth hormone (GH), growth hormone receptor (GHR), transcription factor forkhead box L2 (*FOXL2*) and members of the transforming growth factor beta (TGFβ) superfamily, including growth differentiation factor-9 (*GDF9*), have been intensively studied in chickens [5,20,56,57,58,59]. However, there is no information regarding polymorphisms in the chicken *RAPGEF6* gene. To identify novel DNA markers associated with egg production traits in chickens, we examined polymorphisms in *RAPGEF6* and evaluated their associations with egg production traits in Chinese Jing Hong laying hens.

*RAPGEF6* is a protein-coding gene located on chromosome 13 in chickens, which shows an important role in the reproduction. Previous findings demonstrated that this gene is involved in the directing many vital processes including spermatogenesis [31,32]. This gene also engaged in maintaining a widespread cellular function linked to cellular responses, including cell adhesion, testicular atrophy, substantial decline in sperm quality and dramatic change exhibits in lowering fertility [31,32]. Moreover, *RAPGEF6* gene in mice was demonstrated to be essential for male and/or female fertility. Results of current study showed that the egg-laying rate is significantly associated with three SNPs AX-75745366 (T15836649C), AX-75745340 (A15829057G) and AX-75745341 (A15829303G) which are found in *RAPGEF6* gene. Furthermore, four haplotypes (AAGG, AAAG, AGGG and AGAG) were detected and the association analysis of haplotypes showed that the polymorphisms of SNP AX-75745340 (A15829057G) and AX-75745341 (A15829303G) in *RAPGEF6* gene are significantly associated with egg production traits in Chinese Jing Hong layer chickens. Therefore, these results also strongly support that the three currently identified SNPs that are significantly associated in the *RAPGEF6* gene might serve as a possible potential candidate in molecular genetic markers to aid in the improvement of egg production traits to be used in the breeding technique of chicken. The chi-square test results demonstrated that allelic and genotypic frequencies for the SNPs in the *RAPGEF6* gene were not in Hardy–Weinberg equilibrium (*HWE*). These results suggest that the allelic and genotypic frequencies of the five polymorphic sites in the *RAPGEF6* gene of the Chinese Jing Hong layer chicken population do not remain constant from generation to generation due to the influence of selection, mate choice, migration, and mutation. SNPs were tested and demonstrated a remarkably genetic disequilibrium between alleles, which might change the population structure and genetic drift in the studied populations.

Furthermore, we demonstrated that for SNP-01, AX-75745366 (T15836649C), the frequency of allele T was predominantly higher than allele C and frequency of genotype TT, and TC was higher than that of genotype CC in the Jing Hong layer chicken of P1, P2, and P1 + P2 populations, but only after the whole population was found to exhibit a significant genetic disequilibrium between T and C alleles in *RAPGEF6* (*p* < 0.05). This shows a low genetic diversity in the population, which may mainly be due to the cause of selection. Moreover, gene homozygosity (*Ho*) was higher than gene heterozygosity (*He*) for the SNP-01, AX-75745366 (T15836649C), which was also found to be under genetic disequilibrium. The reason why this phenomenon occurring may be explained mainly by the following two aspects: (i) the mutation of allele T to allele C in this *RAPGEF6* fragment was initially present in the original chicken population at a lower frequency; (ii) this substitution has occurred recently. Even if the allele T was correlated with the significantly higher egg production traits, a breeding purpose to enhance body weight and egg weight traits resulted in the presentation of the allele C, and the selection pressure was not enough to increase the frequency of allele C up to a higher level within a very limited number of generations. If this reason were true, the genetic disequilibrium is easily understood, as mentioned above.

For SNP-03, AX-75745340 (A15829057G), the frequency of allele A is higher than allele G, and the frequency of genotype AA was higher than that of genotype AG in the Jing Hong layer chicken of P1, P2, and in P1 + P2 populations. Although the P1 population showed significant genetic equilibrium (*p* > 0.05), the P2 population did not show genetic equilibrium (*p* < 0.05), and the P1 + P2 population was found to exhibit significant genetic disequilibrium between A and G alleles in *RAPGEF6* (*p* < 0.05). This shows a low genetic diversity in the population, which may be mainly due to the cause of selection. Additionally, gene homozygosity (*Ho*) was higher than heterozygosity (*He*) for the SNP-03: AX-75745340 (A15829057G) was shown to be under genetic disequilibrium. The main reason for this phenomenon may be a tight linkage of the allele A with either an advantageous allele or with an artificially selected and therefore economically desirable trait, such as higher egg production. Egg production traits of Chinese local Jing Hong populations have already been improved to enhance early sexual maturity, egg-laying numbers, and egg weight traits in the last decades. Another reason may be attributed to allele A being naturally one of the dominant alleles during genetic evolution, thus being more conserved and more common than other alleles in this population. It certainly cannot be ignored that the number of birds examined in this population was not enough to demonstrate the true event, and an extreme allele frequency was estimated as a result.

For SNP-05, AX-75745341 (A15829303G), the frequency of allele G was higher than allele A and the frequency of genotype GG was higher than that of genotype AG in the Jing Hong layer chicken of P1, P2, and P1 + P2 populations. Although the P1 population showed genetic equilibrium (*p > 0*.05), the P2 population did not show genetic equilibrium (*p* < 0.05), and the P1 + P2 population was found to exhibit significant genetic disequilibrium between A and G alleles in *RAPGEF6* (*p* < 0.05). This shows a low genetic diversity in the population, which may be mainly due to the cause of selection. Moreover, gene homozygosity (*Ho*) was higher than gene heterozygosity (*He*) for the SNP-05, AX-75745341 (A15829303G). This SNP was found to be under genetic disequilibrium, possibly due to allele G in *RAPGEF6* being naturally one of the predominant alleles during genetic evolution and thus being more conserved and more common than other alleles in this population. Furthermore, it is possible that allele A or G may be tightly linked with either an advantageous allele or with an artificially selected economically favorable trait, such as higher egg production and egg weight. Hence, the homozygotes with genotype AA or GG were either promoted under natural selection pressures to be better adapted, or they were artificially selected for favorable agricultural attributes. In fact, the egg production traits of Chinese local Jing Hong populations have already been improved with the aim to enhance early sexual maturity, egg-laying numbers, and egg weight traits for the past six generations. In this local Chinese Jing Hong chicken breeding, the A or G alleles may be coincidently linked with one or more of the selected breeding traits, thus presenting a possible explanation for higher allelic frequencies. Additionally, it cannot be ignored that the number of birds examined in each population was not enough to demonstrate the true event, and an extreme allele frequency was estimated as a result.

The results of the association study and the linkage disequilibrium (LD) analysis revealed that the three significantly associated SNPs were closely linked together in this region. Linkage disequilibrium (LD) plays a vital role in mapping genes that affect complex diseases and identifying association among genetic markers and functional genes [60]. Understanding LD among SNP also avoids redundant inferences involving non-independent genetic markers. The result of this study indicates that three variants in *RAPGEF6* gene are in significant LD with each other. In the P1 population, two variants, AX-75745341A/G and AX-75745341A/G, showed non-significant LD with each other with low D’ (D’ = 13) and spanning 0 kb in block 1 only. In the case of the P2 population, three variants, AX-75745340A/G, AX-75745341A/G, and AX-75745366T/C, showed significant LD with each other with highly strong D’ (D’ = 93 and 83) and spanning 7 kb in block 1 only. Finally, in the case of the whole P1 + P2 population, two variants, AX-75745340A/G and AX-75735341/G, showed significant LD with each other with strong D’ (D’ = 69) and spanning 0 kb in block 1 only. The LD analysis revealed that three variants, AX-75745340A/G, AX-75745341A/G, and AX-75745366T/C, in the *RAPGEF6* gene were significantly associated with the egg-laying rate.

Collectively, the results of the present study strongly suggest that the three novel SNPs are associated with the egg-laying rate and are thus potential molecular markers for egg productivity in local Chinese Jing Hong chicken breeding.

## 5. Conclusions

In conclusion, five SNPs that are associated with egg-laying rate were selected by the GWAS in a parent population. Out of them, three SNPs were confirmed in two commercial populations with PCR-RFLP genotyping data and egg-laying rates. Based on the results of an association analysis, it has been shown that the T allele at SNP AX-75745366 (t15846449T>C), the A allele at SNP AX-75745340 (a15829057A>G), and the G allele at SNP AX-75745341 (g15829303A>G) in the *RAPGEF6* gene are the most potential candidate molecular genetic markers that can be used to improve the egg production traits in MAS programs. Furthermore, the *RAPGEF6* gene is not only beneficial in males but can also be considered as one of the novel potential candidate gene in regulating egg production traits in females. Our present study therefore brings new insight on the *RAPGEF6* gene function in Chinese Jing Hong chickens. This study not only provides the candidate genetic markers for a marker-assisted selection of Chinese Jing Hong hens, but also provides a basic knowledge for further studies on SNP detection on the *RAPGEF6* genes in other chicken breeds and other animal species.

## 6. Patent

Patent applying No: 201810428916.5 resulting from the work.

## Figures and Tables

**Figure 1 genes-10-00384-f001:**
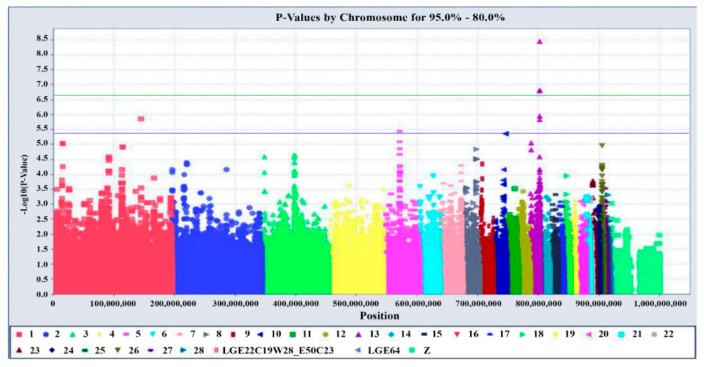
The Manhattan plot shows the association of single nucleotide polymorphisms with egg production traits. Each dot represents one SNP. The figure illustrated the level of statistical significance (y-axis), as measured by the negative log of the corresponding *p* value for each single nucleotide polymorphism (SNP). Each typed SNPs indicate dots of different colors, which are arranged by chromosomal location (x-axis). Imputation was performed on chromosome 13 using only the data of 120 genomes. The highlighted green line indicates the threshold of 5% Bonferroni genome-wide significance (*p* = 2.09 × 10^−7^), and the underlined blue line shows the limit of symbolic genome-wide significance (*p* = 4.18 × 10^−6^), respectively.

**Figure 2 genes-10-00384-f002:**
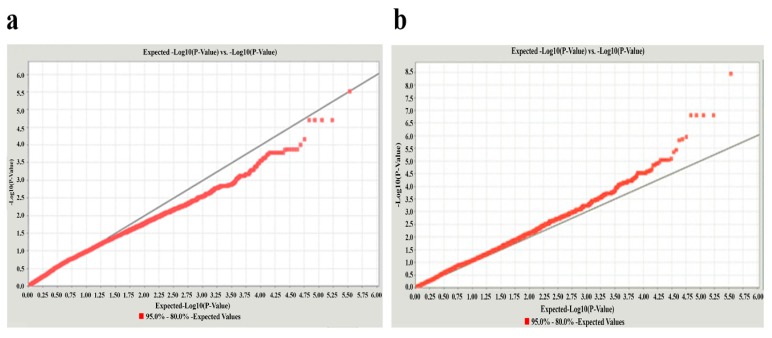
Q-Q (Quantile–Quantile) plots for quality control check and visualizing crude association. (**a**) Unadjusted model; λ = 1.0142 and (**b**) Adjusted model for PCs, Age and Sex; λ = 1.0032. The plots illustrate the relationship between observed log10 (*p* value) in the y-axis and expected log10 (*p* value) in x-axis test statistics and are used as a tool for visualizing appropriate control of population substructure and the presence of association. The left panel (**a**) is based on an unadjusted model, where the deviation is below expected, while the right panel (**b**) is based on an adjusted model for potential cofounders, which brings the tail closer to the y = x line. The observed extreme statistics are suggestive of an association. Data generally falling on the y = x lines suggest no clear systemic bias. Unstandardized λ’s are reported. PCs: principal components.

**Figure 3 genes-10-00384-f003:**
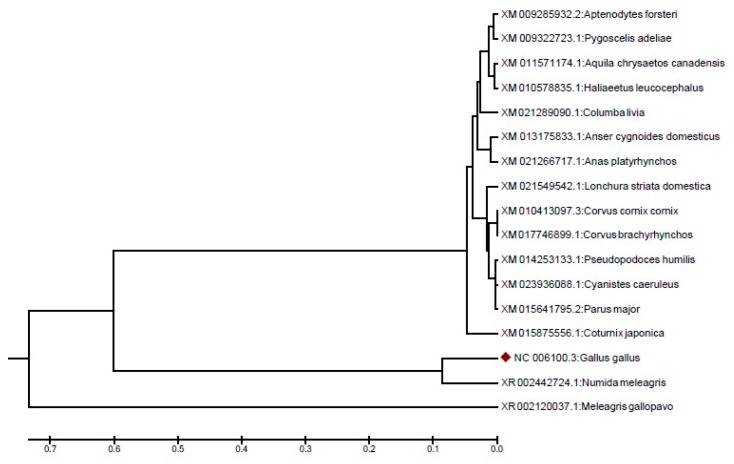
The UPGMA (Unweighted Pair Group Method with Arithmetic Mean) evolutionary relationship tree of chicken (*Gallus gallus*) *RAPGEP6* gene with different species.

**Figure 4 genes-10-00384-f004:**
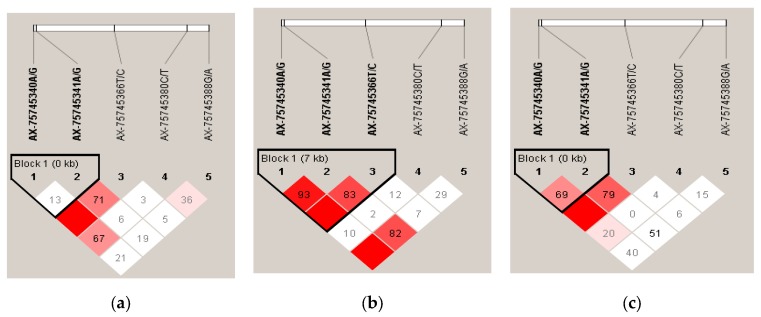
(**a**–**c**) Linkage disequilibrium (LD) of single nucleotide polymorphisms (SNPs) at the *RAPGEF6* gene in egg-laying rate in Chinese Jing Hong Chicken. Pairwise correlation (D’) values are shown between polymorphisms, which were calculated from the genotypic data of 858, 816, and 1676 chicken for P1, P2, and P1 + P2, respectively. Block colors indicate the LD status of SNPs; deep red means high linkages between two SNPs. The haplotypes block was defined by using the default setting of the Haploview software.

**Table 1 genes-10-00384-t001:** Genotypic and allelic frequencies with Hardy–Weinberg equilibrium (HWE) test at the SNP locus of *RAPGEF6* gene in the Chinese Jing Hong chicken population.

SNPs (Location)	Population	*n*	Genotype Frequency			Allele Frequency		χ^2^ (HWE)	*p* Value
SNP-01 (AX-75745366)			TT	TC	CC	T	C		
	P1	858	0.33 (282)	0.52 (450)	0.15 (126)	0.59 (1014)	0.41 (702)	6.17 *	0.0130
	P2	818	0.32 (265)	0.54 (438)	0.14 (115)	0.59 (968)	0.41 (668)	9.57 *	0.0020
	P1 + P2	1676	0.33 (547)	0.53 (888)	0.14 (241)	0.59 (1982)	0.41 (1370)	15.51 *	0.0001
SNP-02 (AX-75745380)			CC	CT	TT	C	T		
	P1	858	0.81 (697)	0.19 (161)	-	0.91 (1555)	0.09 (161)	9.20 *	0.0024
	P2	818	0.82 (668)	0.18 (150)	-	0.91 (1486)	0.09 (150)	8.33 *	0.0039
	P1 + P2	1676	0.81 (1365)	0.19 (311)	-	0.91 (3041)	0.09 (311)	17.53 *	0.0000
SNP-03 (AX-75745340)			AA	AG	GG	A	G		
	P1	858	0.91 (782)	0.09 (76)	-	0.96 (1640)	0.04 (76)	1.84 ^NS^	0.1746
	P2	818	0.78 (642)	0.22 (176)	-	0.89 (1460)	0.11 (176)	11.89 *	0.0006
	P1 + P2	1676	0.85 (1424)	0.15 (252)	-	0.92 (3100)	0.08 (252)	11.08 *	0.0009
SNP-04 (AX-75745388)			GG	GA	AA	G	A		
	P1	858	0.49 (421)	0.51 (437)	-	0.75 (1279)	0.25 (437)	100.16 *	0.0000
	P2	818	0.50 (405)	0.50 (413)	-	0.75 (1223)	0.25 (413)	93.28 *	0.0000
	P1+P2	1676	0.49 (826)	0.51 (850)	-	0.75 (2502)	0.25 (850)	193.44 *	0.0000
SNP-05 (AX-75745341)			AA	AG	GG	A	G		
	P1	858	-	0.12 (104)	0.88 (754)	0.06 (104)	0.94 (1612)	3.57 ^NS^	0.0588
	P2	818	-	0.27 (223)	0.73 (595)	0.14 (223)	0.86 (1413)	20.37 *	0.0000
	P1 + P2	1676	-	0.20 (327)	0.80 (1349)	0.10 (327)	0.90 (3025)	19.58 *	0.0000

P1 = First population, P2 = Second population, P1 + P2 = Both first and second population, *n* = Number of individual, SNP = Single nucleotide polymorphism, RAPGEF6 = Rap Guanine Nucleotide Exchange Factor 6, * *p* < 0.05 was accepted to be statistically significant when the data were analyzed using a Pearson’s goodness-of-fit chi-square test (degree of freedom = 1), NS = Non-significant at *p* ≥ 0.5.

**Table 2 genes-10-00384-t002:** Haplotypes inferred based on the three SNPs and frequencies in *RAPGEF6* gene of Chinese Jing Hong chicken population.

Haplotype	Polymorphism Site			Frequency in Population		
	T15836649C	A15829057G	A15829303G	P1	P2	(P1 + P2)
H1	T	A	G	0.5378	0.4762	0.5089
H2	C	A	G	0.3829	0.3467	0.3644
H3	T	A	A	0.0270	0.0377	0.0322
H4	T	G	A	0.0167	0.0480	0.0316
H5	T	G	G	0.0095	0.0298	0.0186
H6	C	G	G	0.0093	0.0109	0.0106
H7	C	G	A	0.0087	0.0189	0.0144
H8	C	A	A	0.0081	0.0318	0.0193

Haplotype: H1 (TAG), H2 (CAG), H3 (TAA), H4 (TGA), H5 (TGG), H6 (CGG), H7 (CGA), H8 CAA).

**Table 3 genes-10-00384-t003:** Polymorphism information analysis of the SNPs in *RAPGEF6* gene in the Chinese local Jing Hong chicken population.

SNPs (Location)	Population	Number of Chickens (*n*)	Gene Homozygosity (Ho)	Gene Heterozygosity (He)	Effective Allele Number (Ne)	Polymorphism Information Content (PIC)
SNP-01 (AX-75745366)	P1	858	0.5165	0.4835	1.9360	0.3666
	P2	818	0.5168	0.4832	1.9349	0.3665
	P1 + P2	1676	0.5167	0.4833	1.9355	0.3665
SNP-02 (AX-75745380)	P1	858	0.8300	0.1700	1.2049	0.1556
	P2	818	0.8334	0.1666	1.1998	0.1527
	P1 + P2	1676	0.8317	0.1683	1.2024	0.1542
SNP-03 (AX-75745340)	P1	858	0.9153	0.0847	1.0925	0.0811
	P2	818	0.8080	0.1920	1.2376	0.1736
	P1 + P2	1676	0.8609	0.1391	1.1615	0.1294
SNP-04 (AX-75745388)	P1	858	0.6204	0.3796	1.6119	0.3076
	P2	818	0.6226	0.3774	1.6063	0.3062
	P1 + P2	1676	0.6214	0.3786	1.6092	0.3069
SNP-05 (AX-75745341)	P1	858	0.8861	0.1139	1.1285	0.1074
	P2	818	0.7645	0.2355	1.3080	0.2077
	P1 + P2	1676	0.8239	0.1761	1.2137	0.1606

P1 = First population, P2 = Second population, P1 + P2 = Both first and second population.

**Table 4 genes-10-00384-t004:** Association analysis between the genotypes of five polymorphisms in chicken *RAPGEF6* gene and egg-laying rates in the Chinese Jing Hong chicken population.

SNPs (Location)	Population	*n*	Genotype Frequency (LSM ± SEM)			*F*-value, *p* Value and Level of Significance		
						*F*-Value	*p* Value	Level of Significance
SNP-01 (AX-75745366)			TT	TC	CC			
	P1	858	78.63 ± 1.08 ^a^ (282)	80.50 ± 0.86 ^a^ (450)	72.61 ± 1.62 ^b^ (126)	9.25	<0.0001	***
	P2	818	85.30 ± 1.02 ^a^ (265)	80.49 ± 0.79 ^b^ (438)	76.40 ± 1.54 ^c^ (115)	13.29	<0.0001	***
	P1 + P2	1676	81.86 ± 0.75 ^a^ (547)	80.49 ± 0.59 ^a^ (888)	74.42 ± 1.13 ^b^ (241)	15.67	<0.0001	***
SNP-02 (AX-75745380)			CC	CT	TT			
	P1	858	79.20 ± 0.70 (697)	76.67 ± 1.45 (161)	-	2.49	0.1151	NS
	P2	818	81.18 ± 0.65 (668)	82.78 ± 1.37 (150)	-	1.12	0.2896	NS
	P1 + P2	1676	80.17 ± 0.48 (1365)	79.62 ± 1.00 (311)	-	0.25	0.6197	NS
SNP-03 (AX-75745340)			AA	AG	GG			
	P1	858	79.99 ± 0.64 ^a^ (782)	65.69 ± 2.06 ^b^ (76)	-	44.03	<0.0001	***
	P2	818	83.28 ± 0.65 ^a^ (642)	74.87 ± 1.24 ^b^ (176)	-	36.30	<0.0001	***
	P1 + P2	1676	81.47 ± 0.46 ^a^ (1424)	72.10 ± 1.09 ^b^ (252)	-	62.55	<0.0001	***
SNP-04 (AX-75745388)			GG	GA	GG			
	P1	858	78.92 ± 0.90 (421)	78.54 ± 0.88 (437)	-	0.09	0.7633	NS
	P2	818	81.08 ± 0.83 (405)	81.86 ± 0.83 (413)	-	0.45	0.5015	NS
	P1 + P2	1676	79.97 ± 0.61 (826)	80.15 ± 0.61 (850)	-	0.04	0.8355	NS
SNP-05 (AX-75745341)			AA	AG	GG			
	P1	858	-	72.43 ± 1.79 ^b^ (104)	79.59 ± 0.66 ^a^ (754)	14.09	<0.0001	***
	P2	818	-	74.50 ± 1.09 ^b^ (223)	84.09 ± 0.67 ^a^ (595)	56.57	<0.0001	***
	P1 + P2	1676	-	73.84 ± 0.96 ^b^ (327)	81.57 ± 0.47 ^a^ (1349)	51.96	<0.0001	***

P1 = First population, P2 = Second population, P1 + P2 = Total (both first and second) population, *n* = Number of individual, SNP = Single nucleotide polymorphism, *RAPGEF6* = Rap Guanine Nucleotide Exchange Factor 6, LSM = Least squares of mean, SEM = Standard error of mean, G = Genotypes (i.e., TT, TC, CC). The values in parentheses indicate the numbers of chicken for their corresponding group. * = Significant at *p* < 0.1, ** = Significant at *p* < 0.01, *** = Significant at *p* < 0.001, ^a, b, c^ = LSM values bearing different letters in each column are significantly different at *, ** and ***, NS = Non-significant at *p* > 0.5.

**Table 5 genes-10-00384-t005:** Association analysis between the haplotype inferred through AX-75745340 (A15829057G) & AX-75745341 (A15829303G) in chicken *RAPGEF6* gene and egg-laying rate of Chinese Jing Hong chicken population.

Trait	Population	*n*	Haplotype Frequency (LSM ± SEM)				*F*-Value, *p* Value and Level of Significance		
							*F*-Value	*p* Value	Level of Significance
Laying Rate			*H1* (AAGG)	*H2* (AAAG)	*H3* (AGGG)	*H4* (AGAG)			
	P1	858	79.79 ± 0.40 ^ab^ (724)	83.03 ± 2.33 ^a^ (58)	75.85 ± 3.24 ^b^ (30)	59.07 ± 2.62 ^c^ (46)	20.963	<0.0001	***
	P2	818	84.50 ± 0.70 ^a^ (534)	77.27 ± 1.56 ^b^ (108)	80.46 ± 2.07 ^ab^ (61)	71.90 ± 1.51 ^c^ (115)	22.226	<0.0001	***
	P1 + P2	1676	81.76 ± 0.49 ^a^ (1258)	79.28 ± 1.34 ^a^ (166)	78.94 ± 1.8 ^a^ (91)	68.23 ± 1.36 ^b^ (161)	0.246	<0.0001	***

P1 = First population, P2 = Second population, P1 + P2 = Total (both first and second) population, *n* = Number of individual, SNP = Single nucleotide polymorphism, *RAPGEF6* = Rap Guanine Nucleotide Exchange Factor 6, LSM = Least squares of mean, SEM = Standard error of mean, H = Haplotypes (i.e., AAGG, AAAG, AGGG and AGAG). The values in parentheses indicate the number of chicken for their corresponding group. * = Significant at *p* < 0.1, ** = Significant at *p* < 0.01, *** = Significant at *p* < 0.001, ^a, b, c^ = LSM values within a row for each haplotype lacking a common superscript differ significantly *p* < 0.05).

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
