# Peer review of "Novel Polymorphisms in RAPGEF6 Gene Associated with Egg-Laying Rate in Chinese Jing Hong Chicken using Genome-Wide SNP Scan"

_genes, 2019, doi:10.3390/genes10050384_

Reviewer 1 Report

This manuscript describes the identification of novel polymorphisms within the chicken RAPGEF6 gene that are associated with egg laying rate in the Chinese Jing Hong Chicken breed. The science is novel and the observations are of interest to the egg laying industry and scientists that study egg production. In my opinion the rigor of the science and experimental design is sound. 

The manuscript does require editing with respect to English and grammar. I started to note specific edits that require correction, however they became too numerous for me to continue. This did not interfere with my ability to understand the science and interpret the results and I congratulate the Authors on this. My recommendation is that the Authors work closely with someone to re-edit the manuscript to correct the English flaws as it is not publishable in the current state.

Author Response

Thank you for your valuable comments and suggestions on our manuscript. Our response are as follows:

Research Title: A Genome-Wide SNP Scan Reveals Novel Polymorphisms in RAPGEF6 Gene Associated with Egg Laying Rate in Chinese Jing Hong Chicken

Question 1: The manuscript does require editing with respect to English and grammar. I started to note specific edits that require correction, however they became too numerous for me to continue. This did not interfere with my ability to understand the science and interpret the results and I congratulate the Authors on this. My recommendation is that the Authors work closely with someone to re-edit the manuscript to correct the English flaws as it is not publishable in the current state.

Answer: The whole manuscript is edited thoroughly very carefully. We think the present quality of our manuscript might be belongs to the standard of articles published in Genes. So, if you would kind enough to consider and accept our manuscript for publication in your well reputed journal “Genes”, we would remain grateful to you.

Reviewer 2 Report

The Authors reported the association of three SNPs in RAPGEF6 gene with the egg laying rate during the later stage of egg production. They suggest the possible use of these SNPs as potential candidate molecular genetic markers for MAS for egg production traits in chicken breeding.

The topic of the paper is interesting for the readers.

 Line: 160 to 162: how the 120 subjects for GWAS were chosen?

Line 167 to 168: a reference for R package is needed.

Line 169: which software was used? Specify.

Line 221 to 226: Please remove the formula, as it is not necessary!

Line 260 to 270: verify the format of the legend. Improve the quality of the figure1. The green and blue lines cited in the legend are missing and the same for linkage groups (LGE1 and LGE2).

Line 271: Improve the quality of the figure2.

Line 278: “…..potential confounders,…” replace with “…..potential cofounders,…”.

Line 301 to 305: the definition of the SNPs reported in the sentence is different respect to those reported in the table 1 and table 3, please modify as correctly reported in table 4.

Line 475 to 481: “….The RAPGEF6 is a protein coding gene located on chromosome 13 in chickens, which shows an important role in the reproduction. Previous findings demonstrated that this gene is involved in the directing many vital processes including spermatogenesis [31,32]. This gene also engaged in maintaining a widespread cellular function linked to cellular responses, including cell adhesion, testicular atrophy, substantial decline in sperm quality and dramatic change exhibits in lowering fertility [31,32]. Moreover, RAPGEF6 gene in mice was demonstrated to be essential for male and/or female fertility.” move these sentences before line 467.

Line 481: “…..In addition, Chi-square test results demonstrated……not in Hardy-Weinberg equilibrium (HWE).” Modify in : “…..The Chi-square test results demonstrated……not in Hardy-Weinberg equilibrium (HWE).”

Author Response

Thank you for your valuable comments and suggestions on our manuscript. Our response are as follows:

Research Title: A Genome-Wide SNP Scan Reveals Novel Polymorphisms in RAPGEF6 Gene Associated with Egg Laying Rate in Chinese Jing Hong Chicken

1.      Line: 160 to 162: how the 120 subjects for GWAS were chosen?

Answer:  Jing Hong parent hens with extreme performance differences were used to detect SNP markers associated with egg laying rate using GWAS. A high-density SNP array was employed herein to identify associated variants underlying egg production traits using the chicken 600K Affymetrix Axiom HD SNP-array to investigate whether the effects of these QTL is associated with egg production traits[1]. GWAS was performed using a total of 120 Jing Hong parent chicken from among a total of 858 hens of first population. The chicken of first sixty and the last sixty considering egg production trait were used to do GWAS. The first sixty and the last sixty were considered as control and case group respectively.

2.      Line 167 to 168: a reference for R package is needed.

Answer: The reference of the R software is now incorporated. The global view of P-values for all SNP markers was visualized by a Manhattan and Q-Q (Quantile-quantile) plot that was constructed for each trait using the “CM plot” package in the R software (version 3.5)[2].

Reference:

Wang, D.; Sun, Y.; Stang, P.; Berlin, J.A.; Wilcox, M.A.; Li, Q. Comparison of methods for correcting population stratification in a genome-wide association study of rheumatoid arthritis: principal-component analysis versus multidimensional scaling. BMC proc., 2009, 3 (Suppl 7), S109-S109.

 3.      Line 169: which software was used? Specify.

Answer: The name of software is incorporated and the text is modified as: Genome-wide association study analysis were performed using the compressed mixed linear model[3] and carried out using the TASSEL software package (version 5.0)[4]. The genome-wide significance P-value threshold was determined by the “LD adjusted” Boneferroni method[5].

References:

Zhang, Z.; Ersoz, E.; Lai, C.-Q.; Todhunter, R.J.; Tiwari, H.K.; Gore, M.A.; Bradbury, P.J.; Yu, J.; Arnett, D.K.; Ordovas, J.M., et al. Mixed linear model approach adapted for genome-wide association studies. Nat. Genet., 2010, 42, 355, doi:10.1038/ng.546.

Bradbury, P.J.; Zhang, Z.; Kroon, D.E.; Casstevens, T.M.; Ramdoss, Y.; Buckler, E.S. TASSEL: software for association mapping of complex traits in diverse samples. Bioinformatics, 2007, 23(1), 2633-2635, doi:10.1093/bioinformatics/btm308.

Duggal, P.; Gillanders, E.M.; Holmes, T.N.; Bailey-Wilson, J.E. Establishing an adjusted p-value threshold to control the family-wide type 1 error in genome wide association studies. BMC Genomics, 2008, 9, 516-516, doi:10.1186/1471-2164-9-516.

4.      Line 221 to 226: Please remove the formula, as it is not necessary!

Answer: The formula of gene frequency and genotype frequency used in between the line 221 to 226 has been removed from text as directed.

 5.      Line 260 to 270: verify the format of the legend. Improve the quality of the figure1. The green and blue lines cited in the legend are missing and the same for linkage groups (LGE1 and LGE2).

Answer: The format of the figure legend has been corrected, the quality of the figure 1 has been improved incorporating new one and the text of the legend has been checked and corrected according to reviewer’s comments.

 6.      Line 271: Improve the quality of the figure 2.

Answer: The quality of the figure 2 has been improved incorporating new one according to reviewer’s comments.

 7.      Line 278: “…..potential confounders,…” replace with “…..potential cofounders,…”.

Answer: In Line 278: The word potential confounders has been replaced by the word cofounders.

 8.      Line 301 to 305: the definition of the SNPs reported in the sentence is different respect to those reported in the table 1 and table 3, please modify as correctly reported in table 4.

Answer: SNP definition in line between 301 to 305 as reported in sentence has been respectfully accepted the reviewer’s comment and has been modified in table 1 and 3 following as the table 4.

9.      Line 475 to 481: “…. The RAPGEF6 is a protein coding gene located on chromosome 13 in chickens, which shows an important role in the reproduction. Previous findings demonstrated that this gene is involved in the directing many vital processes including spermatogenesis [31,32]. This gene also engaged in maintaining a widespread cellular function linked to cellular responses, including cell adhesion, testicular atrophy, substantial decline in sperm quality and dramatic change exhibits in lowering fertility [31,32]. Moreover, RAPGEF6 gene in mice was demonstrated to be essential for male and/or female fertility.” move these sentences before line 467.

Answer: The Line 475 to 481 has been moved before the line 467 according to the reviewer’s comments.

10.  Line 481: “…..In addition, Chi-square test results demonstrated……not in Hardy-Weinberg equilibrium (HWE).” Modify in : “…..The Chi-square test results demonstrated……not in Hardy-Weinberg equilibrium (HWE).”

Answer: Line 481: The sentence has been corrected according to the reviewer’s comments.

Round  2

Reviewer 2 Report

The manuscript has been significantly improved and no other revision are needed.